

# Detrimental effects of heat stress on grain weight and quality in rice (*Oryza sativa* L.) are aggravated by decreased relative humidity

Haoliang Yan, Chunhu Wang, Ke Liu and Xiaohai Tian

Hubei Collaborative Innovation Center for Grain Industry/College of Agriculture, Yangtze University, Jingzhou Hubei, China

## ABSTRACT

There is concern over the impact of global warming on rice production due increased heat stress, coupled with decreased relative humidity (RH). It is unknown how rice yield and quality are affected by heat stress and decreased RH during the grain filling stage. We conducted experiments in controlled growth chambers on six rice cultivars, varying in heat tolerance using 12 combinative treatments of three factors: two RH levels (75% and 85%), three temperature levels (the daily maximum temperature at 33 °C, 35 °C, and 37 °C), and two durations (8 d and 15 d after anthesis). Results showed that RH75% with temperature treatments significantly reduced grain weight, which was higher than RH85%. The same trend was also observed for both head rice rate and chalkiness. R168 was the most heat-tolerant cultivar, but it still had some differences in grain weight, head rice rate, and chalkiness between the two RH regimes. The lower RH was most detrimental at 35 °C, and to a lesser extent at 33 °C, but had a negligible effect at 37 °C. Our results provide a better understanding of temperature and RH's interaction effects on rice quality during the grain filling stage, suggesting that RH should be considered in heat tolerance screening and identification to facilitate rice breeding and genetic improvement.

## INTRODUCTION

The average global surface temperature has risen by an estimated 0.85 °C from 1880–2012 (*Stocker et al., 2013*). Evaporation over land has increased because the rate of warming is greater on land than in the sea. However, the limitations of soil moisture, water supply, and crop transpiration cause a decrease in the near-surface relative humidity (RH) over land. As a result, most land types, except coastal areas, will become drier (*Byrne & O'Gorman, 2016*; *Dai, Zhao & Chen, 2018*; *Orimoloye et al., 2018*; *Po-Chedley et al., 2018*).

Rice is one of the most important staple cereals, providing food for more than half of the world's population (*Seck et al., 2012*). However, rice yields fluctuate considerably and are susceptible to climate change (*Jagadish, Murty & Quick, 2015*; *Yan et al., 2017*).

Corresponding author
Xiaohai Tian, xiaohait@sina.com

With each 1 °C increment in the whole-season minimum temperature, there is a 10% loss in yields (*Peng et al., 2004*). In addition to the detrimental effects of heat stress, heat can also damage the quality of the grain, leading to considerable economic losses (*Lyman et al., 2013*).

The grain filling stage is one of the most critical periods for rice yield and grain quality formation and is sensitive to environmental stress (*Yoshida, 1981*). High temperatures during this period result in an accelerated grain filling rate, a shortened grain filling duration, lowered grain weight, and/or deteriorated milling quality caused by increased amounts of chalky and fissured grains forced into maturity by higher temperatures (*Ambardekar et al., 2011*; *Bao, 2019*; *Cooper et al., 2006*; *Zhou, Yun & He, 2019*). The early and middle grain-filling periods are the most susceptible to heat stressors, especially during the first 15 days (*Cooper, Siebenmorgen & Counce, 2008*; *Cooper et al., 2006*; *Wu, Chang & Lur, 2016*). It has been suggested that a temperature range of 25–29 °C is optimal for the first 15–20 d of the grain-filling stage, but it is subject to the response of different cultivars with their varietal tolerance (*Abayawickrama et al., 2017*; *Morita, Wada & Matsue, 2016*; *Wu, Chang & Lur, 2016*).

Starch biosynthesis is inhibited at high temperatures in the developing grains (*Yamakawa & Hakata, 2010*; *Yamakawa et al., 2007*), and many immature starch granules are found in the endosperm cells (*Zakaria et al., 2002*). Altered expressions of α-amylase genes and increased enzymatic activity were detected, suggesting that starch was degraded under elevated temperatures (*Hakata et al., 2012*). Sucrose transport and metabolism in the phloem of the leaf, sheath, stem, and grains are also inhibited in a heat stress-susceptible cultivar in a heat-stressed environment when compared to a heat stress-tolerant cultivar (*Tanamachi et al., 2016*; *Zhang et al., 2018a*). Heat stress may cause a starch shortage, reduced grain weight, and poor quality.

The panicle is a significant determinant for yield stability and grain quality regarding heat-induced damage in grain yield. It is affected by high-temperature stress during the grain filling stage. Physiologically, panicle development is influenced by air temperature and humidity (*Weerakoon, Maruyama & Ohba, 2008*). The lower RH conditions (RH15%) in New South Wales, Australia alleviated heat-induced damage in rice through transpiration cooling during the flowering stage when compared with the humidity in Hubei, China (*Matsui et al., 2014*; *Tian et al., 2010*; *Yoshimoto et al., 2012*). Hot and dry wind conditions can accelerate water loss in panicles and increase the chalky grain rate during the grain filling stage (*Hiroshi et al., 2012*; *Kang et al., 2003*). *Oya & Yoshida (2008)* found a clear difference in chalky formation among varieties with wind treatments. It is not well understood how rice yield and quality are affected by heat stress coupled with different RH during the grain filling stage.

The endosperm cells of rice grains rely on osmotic adjustment to maintain cell turgor under heat stress (*Wada et al., 2019*), similar to rice grown under dry wind (*Wada et al., 2014*, *2011*). This suggests a similar mechanism between the responses to high temperature and dry wind. A lower RH in the air also reduces head rice yield (*Thompson & Mutters, 2006*). In northern India, RH showed a significant decline of 1.44% per year, which significantly affected the yield of most crops (*Chakraborty & Hazari, 2017*).

It is estimated that for every 1% decrease in average RH during the growing season in China, rice yield decreases by 0.75% (*Zhang, Zhang & Chen, 2017*). These studies indicate that RH variation during the grain filling stage may impact rice yield and quality, but whether such a difference is aggravated in cultivars by the interaction of RH with temperature remains to be tested.

In the Jianghan basin of the middle Yangtze River Valley of Central China, mid-season *indica* rice varieties often suffer yield losses due to heat stress (*Tian et al., 2010*). Higher summer temperatures are always accompanied by decreased humidity (*Gong et al., 2006*; *Guo et al., 2016*). A maximum temperature of approximately 35 °C or a daily mean temperature of 30 °C lasting more than 3 days causes RH to decrease from ~85% to ~70% (*Tian et al., 2010*). Climate change has resulted in longer periods of higher temperatures in this region (*Liu et al., 2019*; *Tan & Shen, 2016*). This study hypothesized that high temperatures, coupled with decreased RH, aggravate rice grain quality's adverse effects. We conducted experiments in a controlled environment with six rice cultivars varying in heat stress tolerance. We sought to determine: (1) the effect of RH on high temperature-induced grain weight and quality loss; (2) varietal differences under combinations of RH and temperature treatments.

## MATERIALS AND METHODS

### Plant materials

Six rice cultivars with different heat tolerances at the grain filling stage were sourced from our previous studies. They include three rice hybrids: Liangyou27 (LY27), Liangyou6 (LY6), and Zhuliangyou47 (ZLY47), and three conventional varieties: 16343, R168, and IR64. R168 had a smaller grain weight and quality difference between temperature conditions. IR64 and the other four varieties showed sensitivity to high temperature (Table S1).

### Growth conditions

A pot experiment was conducted in 2017 on Yangtze University experimental farm (Jingzhou City, 112°09′E, 30°21′N, 32 masl) in the western part of the Jianghan Basin in China.

Seeds were sown on April 8, 2017. Twenty-day-old seedlings were transplanted to each plastic pot (inner diameter 30 cm, height 30 cm) containing 12.5 kg soil and 8 g N:P:K compound fertilizer (26:10:15). After transplantation, the soil surface in the pots was kept submerged until maturity in natural conditions. Tillers were cut off after they emerged, leaving only the main stem in each plant throughout the entire growth period.

### Treatment

Six controlled environments with a combined air temperature and relative humidity (RH) level were simulated in a growth chamber (AGC-MR, Zhejiang Qiushi Environment Co., Ltd., Zhejiang, China). Experimental treatments included two RH levels (i.e., RH75% and RH85%), three temperature levels (daily maximum temperatures of 33 °C, 35 °C, and 37 °C), and two durations (8 and 15 days after anthesis) (Fig. 1). Hourly temperature

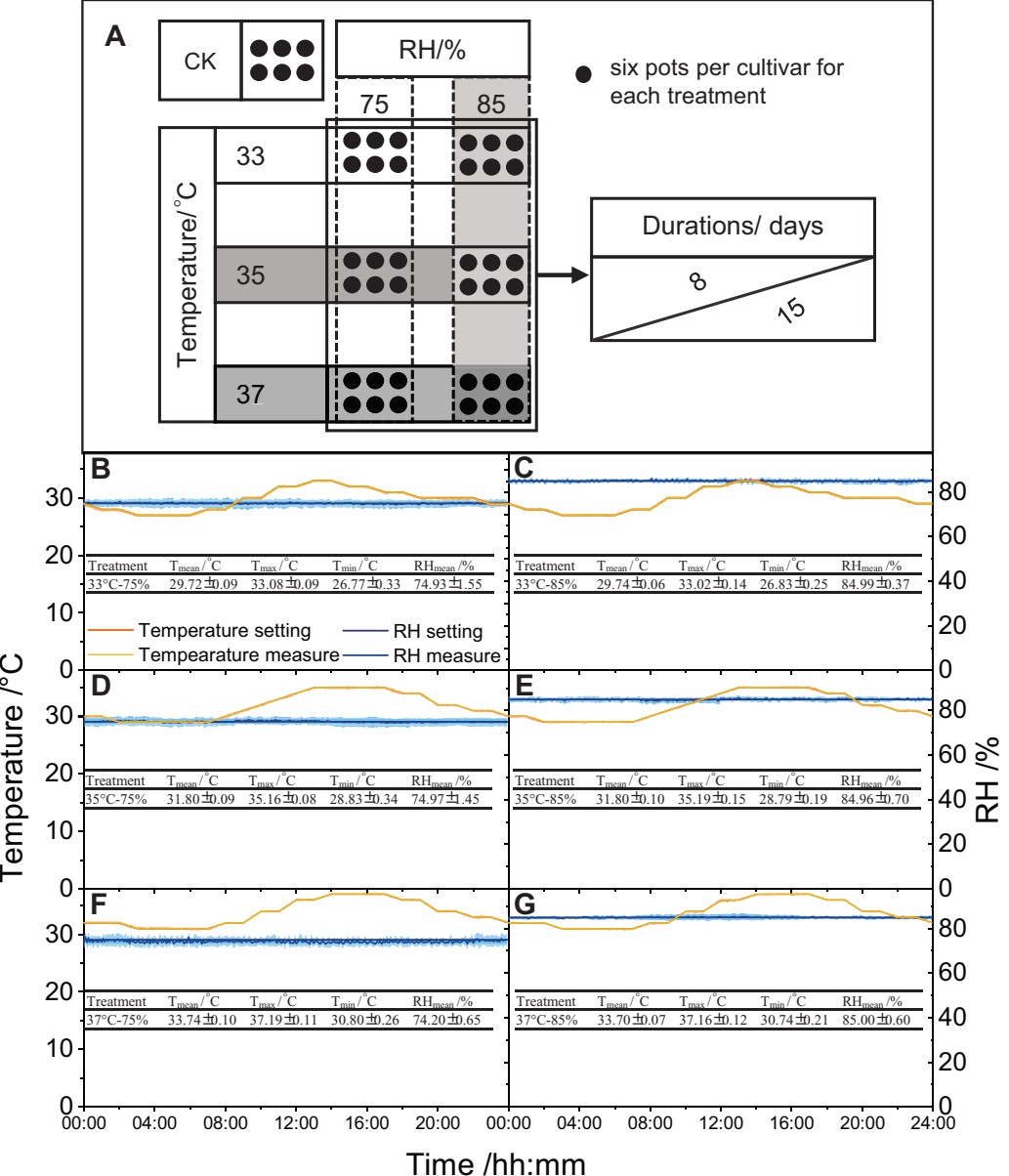

**Figure 1 Diagram of all treatments (A), temperature and RH diurnal variation, maximum temperature, average temperature, minimum temperature, and average RH in each treatment (B–G).** The measured value was the average of 15 d, and the shading around the measured value indicates the standard error of the measured values of 15 d.

changes were recorded daily, and the results are shown in Fig. 1. The RH was kept constant in each treatment environment.

Panicles heading on the same day were marked for treatments after anthesis. Six pots of each cultivar were subjected to each combination of the controlled environment (temperature × RH combinations) for eight and 15 days, respectively. The plants grown under natural conditions (six pots per cultivar) were used as controls. Temperature and RH are shown in Table S2.

## Measurements of grain weight, milling quality, and chalkiness

Filled grains were harvested and sun-dried to a moisture content of 13%. We measured thousand-grain weights with three replications. Samples were stored for 3 months (15–20 °C, RH: 10–20%) and then the milling quality and chalkiness were measured. Rice grain samples (30 g for each cultivar per treatment) were dehulled and polished for 30 s (JDMZ100, Beijing Dongfu Jiuheng Instrument Technology Co., Ltd., Beijing, China) to obtain milled rice. The head rice was then separated and weighed. The weight of the head rice to the sample weight (30 g) was calculated as the head rice rate. The chalkiness of the head rice was evaluated with a rice appearance quality tester (JMWT12, Beijing Dongfu Jiuheng Instrument Technology Co., Ltd., Beijing, China).

## Stress tolerance estimation

Stress tolerance was evaluated by the membership function value (MFV) based on the theory of fuzzy mathematics (*Zadeh, 1965*). We used a method modified from *Chen et al. (2012)* and *Liu, Yang & Hu (2015)*. The heat-tolerant coefficient (HC) was calculated as the ratio of the value in the combinative treatment of temperature × RH × duration to that in the control of the same cultivars for individual traits, using the following equation:

$$HC_{ijk} = \frac{X_{ijk}}{CK_{ij}}$$

where $HC_{ijk}$ is the heat-tolerant coefficient of the trait ($i$) for cultivar ($j$) in treatment ($k$), $X_{ijk}$ is the value of the trait ($i$) for the cultivar ($j$) in the treatment ($k$), $CK_{ij}$ is the value of the trait ($i$) for the cultivar ($j$) under the control condition.

As with HC, MFV for grain weight and head rice rate were calculated, using the following equation:

$$MFV_{ijk} = \frac{HC_{ijk} - Min(HC_i)}{Max(HC_i) - Min(HC_i)}$$

where $MFV_{ijk}$ is the membership function value of heat tolerance of the trait ($i$) for the cultivar ($j$) in treatment ($k$), $HC_{ijk}$ is the same as earlier defined, $HC_i$ is the heat-tolerant coefficient value of the trait ($i$) over all cultivars and all treatments.

The chalkiness value tends to increase after treatment, which is contrary to the changes in grain weight and milled rice rate, and the absolute value of chalkiness under control conditions is usually very small. The MFV for chalkiness was calculated using the following equation:

$$MFV_{ijk} = 1 - \frac{X_{ijk} - Min(X_i)}{Max(X_i) - Min(X_i)}$$

where $X_i$ is the value of the trait ($i$) over all cultivars and all treatments.

For these three traits, MFVs are dimensionless, real number interval [0,1], standing for individual cultivar's heat tolerance under treatments.

## Data analysis

We used Microsoft Excel 2019 for data entry and collating and conducted the analysis of variance (ANOVA) using the R package "agricolae" in R 3.6.0 to determine the effect of treatment factors on grain weight and quality. We performed comparisons between treatment means using the least significant difference test (LSD) at $P \leq 0.05$.

A full model was established via function "lm" in R package "stats" to evaluate the treatment factors and their interactions on grain weight, head rice, and chalkiness. Then, a stepwise backward selection based on the Akaike's information criterion (AIC) which performed with function "stepAIC" in R package "MASS" in R 3.6.0 was used to find out the factors that affected the grain weight, head rice rate, and chalkiness most (*Venables, Ripley & Venables, 2002*). The significance of each predictor was tested via a Student's *t*-test, all treatment factors and their interactions left in the model were significant at the 0.01 level.

The full model of multiple linear regression as follow:

$$Y = \beta_1 X_1 + \beta_2 X_2 + \beta_3 X_3 + \beta_4 X_4 + \beta_{12} X_1 X_2 + \beta_{13} X_1 X_3 + \beta_{23} X_2 X_3 + \beta_{123} X_1 X_2 X_3 + e$$

where $Y$ represents grain weight, head rice rate, and chalkiness in each treatment, respectively; $X_1$ represents the max daily temperature in each treatment; $X_2$ represents the relative humidity in each treatment; $X_3$ represents the treatment duration days in each treatment; and $X_4$ represents the grain weight, head rice rate, and chalkiness of each cultivar under controlled conditions to eliminate the differences in the variety; $\beta_1$, $\beta_2$, $\beta_3$, and $\beta_4$, are linear coefficients, $\beta_{12}$, $\beta_{13}$, $\beta_{23}$, and $\beta_{123}$ represent interaction coefficients, with $e$ representing the intercept.

# RESULTS

## Grain weight

The effects of temperature, RH, and temperature × RH on grain weight were significant among six cultivars (Table 1). The duration of treatment also had a significant effect on grain weight, except for R168. Grain weight was significantly reduced in cultivars at RH 75% coupled temperature treatments, and the reduction was higher than that in RH 85% over the same three temperature treatments (Fig. 2). Under the same temperature and RH conditions, grain weight significantly decreased from 8 d to 15 d of the duration treatment (Fig. 2). The impact of different RH regime treatments (RH75% vs. RH85%) on grain weight was smaller at 37 °C than at 33 °C and 35 °C (Fig. 3).

Multiple regression analysis showed that grain weight decreased when temperature and durations increased, which was the opposite effect when RH increased (Table 2).

MFVs of each trait displayed in Table 3 showed the heat tolerance of each cultivar under treatments, with the mean value as a comprehensive index for the evaluation of heat stress tolerance of each cultivar. R168 was found to have the highest mean MFV (0.70), followed by IR64 (0.55), then LY27, LY6, ZLY47, and 16343 (0.54, 0.50, 0.45, and 0.44, respectively). R168 also showed the smallest difference of MFVs between RH conditions.

**Table 1 Summary of ANOVA for grain weight, head rice rate, and chalkiness in each cultivar.**

| Cultivars | Factor | Grain weight | Head rice rate | Chalkiness |
|---|---|---|---|---|
| LY27 | T | 506.53*** | 81.25*** | 1,296.53*** |
| | RH | 55.35*** | 145.72*** | 60.14*** |
| | D | 77.77*** | 34.40*** | 168.07*** |
| | T×RH | 42.66*** | 21.54*** | 88.60*** |
| | T×D | 3.02 | 3.62* | 7.31** |
| | RH×D | 7.26* | 30.11*** | 8.44** |
| | T×RH×D | 41.25*** | 16.27*** | 73.24*** |
| LY6 | T | 178.54*** | 13.45*** | 1,340.38*** |
| | RH | 188.26*** | 46.68*** | 413.12*** |
| | D | 22.82*** | 61.19*** | 26.03*** |
| | T×RH | 7.45** | 1.41 | 11.26*** |
| | T×D | 2.95 | 2.63 | 2.08 |
| | RH×D | 14.01** | 0.01 | 7.18* |
| | T×RH×D | 0.21 | 1.69 | 11.90*** |
| ZLY47 | T | 23.07*** | 86.81*** | 429.42*** |
| | RH | 13.24** | 23.61*** | 147.61*** |
| | D | 49.20*** | 2.95 | 21.10*** |
| | T×RH | 38.45*** | 1.28 | 50.57*** |
| | T×D | 2.03 | 4.58* | 4.60* |
| | RH×D | 18.33*** | 6.63* | 0.13 |
| | T×RH×D | 0.03 | 3.38 | 1.59 |
| R168 | T | 190.41*** | 23.13*** | 343.66*** |
| | RH | 13.46** | 9.86** | 52.80*** |
| | D | 1.41 | 105.14*** | 87.47*** |
| | T×RH | 4.42* | 4.30* | 30.89*** |
| | T×D | 6.76** | 28.38*** | 0.4 |
| | RH×D | 0.24 | 13.40** | 0.12 |
| | T×RH×D | 3.59* | 18.31*** | 11.08*** |
| IR64 | T | 339.68*** | 244.91*** | 1,541.63*** |
| | RH | 6.84* | 121.39*** | 7.47* |
| | D | 62.03*** | 9.80** | 46.94*** |
| | T×RH | 11.16*** | 1.41 | 22.22*** |
| | T×D | 11.92*** | 9.42*** | 3.91* |
| | RH×D | 0.73 | 40.74*** | 33.28*** |
| | T×RH×D | 11.53*** | 12.01*** | 0.15 |
| 16343 | T | 825.01*** | 437.00*** | 948.33*** |
| | RH | 119.69*** | 216.44*** | 11.84** |
| | D | 285.66*** | 35.24*** | 3.2 |
| | T×RH | 4.36* | 9.57*** | 16.78*** |
| | T×D | 26.94*** | 7.34** | 13.79*** |
| | RH×D | 44.35*** | 4.63* | 0.04 |
| | T×RH×D | 49.06*** | 0.3 | 5.00* |

Note:
Factors: Temperature (T), Relative humidity (RH), Duration days (D). Data are *F*-values with indication of significance levels (* P < 0.05; ** P < 0.01; *** P < 0.001).

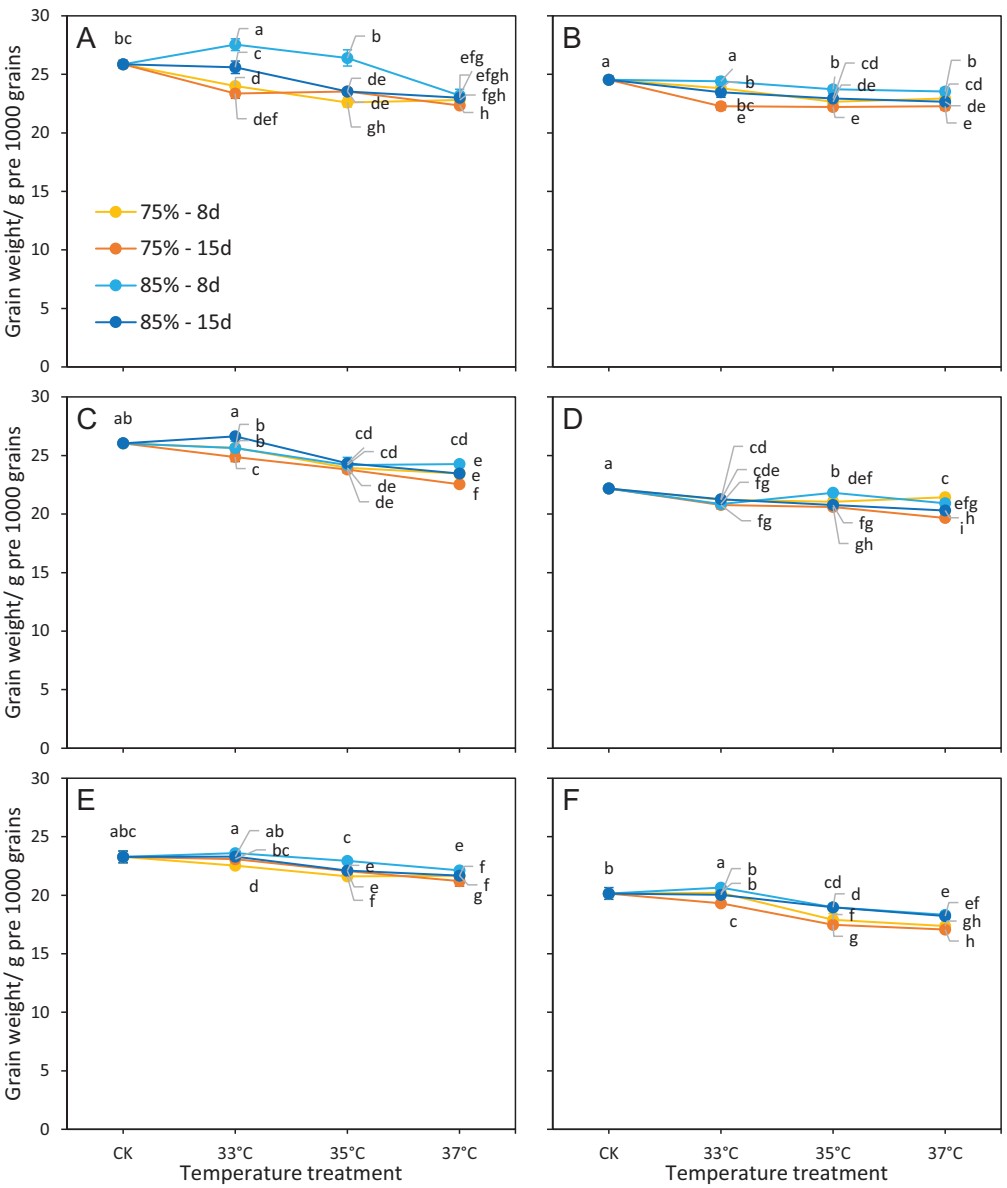

**Figure 2 Grain weight responses to different treatments.** (A) LY27, (B) LY6, (C) ZLY47, (D) R168, (E) IR64, and (F) 16343. Vertical bars denote standard deviations ($n = 3$), a different lowercase letter indicates significant differences among means of different treatments as determined by the LSD test ($p < 0.05$).

The treatment of 35°C by RH75% at 15 days showed the largest mean MFV difference among the cultivars and could be used to evaluate high-temperature tolerance.

R168 and IR64 were also the two cultivars with the highest mean MFVs of grain weight and the smallest difference between RH conditions (Table 3). Their grain weight decreased >2 g at 37 °C (Fig. 2). ZLY47 and 16343 showed high MFVs under temperatures of 33 °C and their grain weight decreased >2 g at 35 °C or 37 °C. LY27 and LY6 showed small MFVs at RH75% over temperature treatments, and their grain weight decreased >2 g at all three temperature treatments.
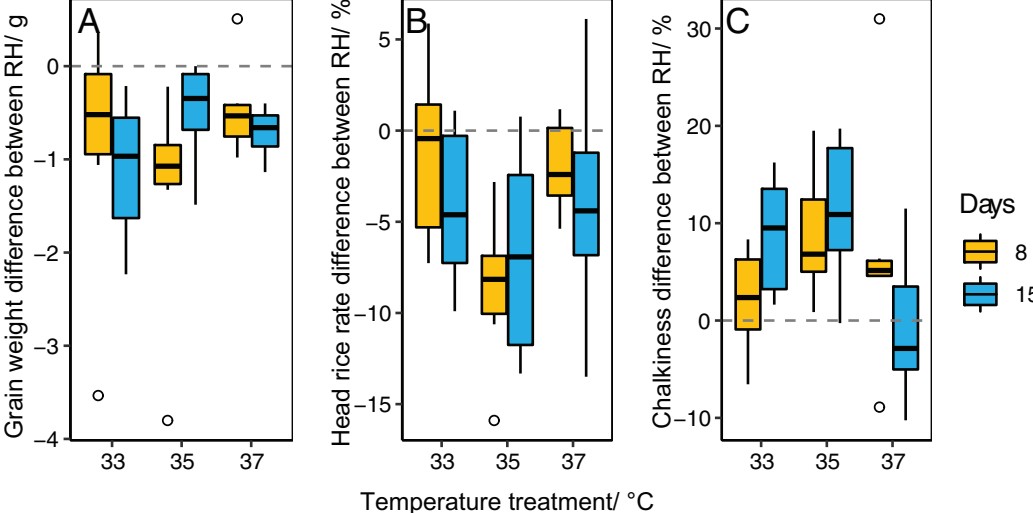

**Figure 3** **Boxplot shows the difference between relative humidity treatments (RH75% vs. RH85%) in different temperature treatments and durations.** (A) Grain weight, (B) head rice rate, and (C) Chalkiness. Data for each boxplot are the value at RH75% minus the value at RH85% under each temperature × duration treatment of six cultivars.

**Table 2 Parameter estimation, *r* square, and *F*-value of multiple regression.**

| Variables | Coefficient | | |
|---|---|---|---|
| | **Grain weight** | **Head rice rate** | **Chalkiness** |
| $X_1$ | −0.3918*** | −6.0261*** | 40.0446** |
| $X_2$ | 0.0837*** | 0.4666*** | 14.9749* |
| $X_3$ | −0.0764*** | −0.4154*** | 123.0023** |
| $X_4$ | 0.9153*** | −0.8432*** | 1.8826*** |
| $X_1X_2$ | – | – | −0.4405** |
| $X_1X_3$ | – | – | −3.4569** |
| $X_2X_3$ | – | – | −1.5463** |
| $X_1X_2X_3$ | – | – | 0.0438** |
| e (intercept) | 8.4565*** | 272.5611*** | −1,361.0483** |
| $R^2$ | 0.8869 | 0.6513 | 0.6989 |
| Adjusted $R^2$ | 0.8862 | 0.6490 | 0.6944 |
| *F*-value | 1,260.52*** | 283.47*** | 155.82*** |

**Note:**
The asterisks near the coefficient value indicates significance at ***($P < 0.001$), **($P < 0.01$), *($P < 0.05$). "-" represent the variable was dropped in the final model. $X_1$: max daily temperature (°C); $X_2$: relative humidity (%); $X_3$: duration days (d); $X_4$: value of each cultivar under controlled conditions.

## Head rice rate

The head rice rate of all cultivars was significantly affected by temperature and RH (Table 1). Durations also showed a significant detrimental effect on the head rice rate except in cultivar ZLY47. Interactive effects of temperature × durations and RH × durations were significant for the head rice rate among cultivars, except LY6. The head rice rate significantly decreased in cultivars at RH75% over temperature treatments, and the

**Table 3 Membership function values (MFVs) of grain weight, head rice rate, and chalkiness for each cultivar in each treatment.**

| Cultivars | Temperature/ °C | RH/ % | Duration days | MFV of grain weight | MFV of head rice rate | MFV of chalkiness | Mean |
|---|---|---|---|---|---|---|---|
| LY27 | 33 | 75 | 8 | 0.37 | 0.89 | 0.82 | 0.70 |
| | | | 15 | 0.26 | 0.79 | 0.63 | 0.56 |
| | | 85 | 8 | 1.00 | 0.88 | 0.74 | 0.87 |
| | | | 15 | 0.66 | 0.87 | 0.73 | 0.75 |
| | 35 | 75 | 8 | 0.12 | 0.68 | 0.60 | 0.47 |
| | | | 15 | 0.29 | 0.49 | 0.49 | 0.42 |
| | | 85 | 8 | 0.80 | 0.74 | 0.71 | 0.75 |
| | | | 15 | 0.29 | 0.74 | 0.75 | 0.59 |
| | 37 | 75 | 8 | 0.16 | 0.50 | 0.41 | 0.35 |
| | | | 15 | 0.07 | 0.49 | 0.40 | 0.32 |
| | | 85 | 8 | 0.23 | 0.54 | 0.49 | 0.42 |
| | | | 15 | 0.19 | 0.36 | 0.32 | 0.29 |
| LY6 | 33 | 75 | 8 | 0.57 | 0.73 | 0.63 | 0.64 |
| | | | 15 | 0.28 | 0.71 | 0.49 | 0.49 |
| | | 85 | 8 | 0.68 | 0.86 | 0.74 | 0.76 |
| | | | 15 | 0.50 | 0.82 | 0.71 | 0.68 |
| | 35 | 75 | 8 | 0.35 | 0.48 | 0.48 | 0.44 |
| | | | 15 | 0.27 | 0.45 | 0.42 | 0.38 |
| | | 85 | 8 | 0.55 | 0.69 | 0.66 | 0.64 |
| | | | 15 | 0.40 | 0.66 | 0.68 | 0.58 |
| | 37 | 75 | 8 | 0.40 | 0.22 | 0.37 | 0.33 |
| | | | 15 | 0.28 | 0.07 | 0.27 | 0.21 |
| | | 85 | 8 | 0.52 | 0.33 | 0.43 | 0.43 |
| | | | 15 | 0.35 | 0.34 | 0.42 | 0.37 |
| ZLY47 | 33 | 75 | 8 | 0.63 | 0.66 | 0.49 | 0.60 |
| | | | 15 | 0.49 | 0.63 | 0.38 | 0.50 |
| | | 85 | 8 | 0.63 | 0.81 | 0.59 | 0.68 |
| | | | 15 | 0.81 | 0.84 | 0.57 | 0.74 |
| | 35 | 75 | 8 | 0.33 | 0.32 | 0.52 | 0.39 |
| | | | 15 | 0.31 | 0.25 | 0.38 | 0.31 |
| | | 85 | 8 | 0.37 | 0.65 | 0.53 | 0.52 |
| | | | 15 | 0.40 | 0.53 | 0.47 | 0.47 |
| | 37 | 75 | 8 | 0.25 | 0.31 | 0.50 | 0.35 |
| | | | 15 | 0.08 | 0.22 | 0.40 | 0.24 |
| | | 85 | 8 | 0.39 | 0.29 | 0.38 | 0.35 |
| | | | 15 | 0.25 | 0.23 | 0.38 | 0.28 |
| R168 | 33 | 75 | 8 | 0.50 | 0.96 | 0.98 | 0.81 |
| | | | 15 | 0.41 | 0.88 | 0.89 | 0.73 |
| | | 85 | 8 | 0.43 | 0.92 | 0.95 | 0.77 |
| | | | 15 | 0.51 | 0.85 | 0.92 | 0.76 |

| Cultivars | Temperature/ °C | RH/ % | Duration days | MFV of grain weight | MFV of head rice rate | MFV of chalkiness | Mean |
|---|---|---|---|---|---|---|---|
| | 35 | 75 | 8 | 0.47 | 0.82 | 0.89 | 0.73 |
| | | | 15 | 0.38 | 0.80 | 0.93 | 0.70 |
| | | 85 | 8 | 0.63 | 0.96 | 0.96 | 0.85 |
| | | | 15 | 0.41 | 0.85 | 0.92 | 0.73 |
| | 37 | 75 | 8 | 0.55 | 0.67 | 0.68 | 0.63 |
| | | | 15 | 0.18 | 0.53 | 0.71 | 0.47 |
| | | 85 | 8 | 0.44 | 0.73 | 0.75 | 0.64 |
| | | | 15 | 0.32 | 0.70 | 0.78 | 0.60 |
| IR64 | 33 | 75 | 8 | 0.55 | 0.80 | 0.96 | 0.77 |
| | | | 15 | 0.66 | 0.71 | 0.94 | 0.77 |
| | | 85 | 8 | 0.76 | 0.83 | 1.00 | 0.86 |
| | | | 15 | 0.71 | 0.86 | 0.96 | 0.84 |
| | 35 | 75 | 8 | 0.37 | 0.27 | 0.74 | 0.46 |
| | | | 15 | 0.46 | 0.29 | 0.55 | 0.43 |
| | | 85 | 8 | 0.63 | 0.43 | 0.80 | 0.62 |
| | | | 15 | 0.47 | 0.27 | 0.73 | 0.49 |
| | 37 | 75 | 8 | 0.38 | 0.17 | 0.61 | 0.39 |
| | | | 15 | 0.29 | 0.00 | 0.52 | 0.27 |
| | | 85 | 8 | 0.47 | 0.15 | 0.67 | 0.43 |
| | | | 15 | 0.39 | 0.10 | 0.39 | 0.29 |
| 16343 | 33 | 75 | 8 | 0.72 | 0.97 | 0.76 | 0.82 |
| | | | 15 | 0.51 | 1.00 | 0.61 | 0.71 |
| | | 85 | 8 | 0.82 | 0.85 | 0.78 | 0.82 |
| | | | 15 | 0.68 | 0.98 | 0.77 | 0.81 |
| | 35 | 75 | 8 | 0.19 | 0.25 | 0.39 | 0.28 |
| | | | 15 | 0.09 | 0.26 | 0.21 | 0.19 |
| | | 85 | 8 | 0.44 | 0.43 | 0.65 | 0.51 |
| | | | 15 | 0.43 | 0.33 | 0.32 | 0.36 |
| | 37 | 75 | 8 | 0.06 | 0.28 | 0.11 | 0.15 |
| | | | 15 | 0.00 | 0.19 | 0.05 | 0.08 |
| | | 85 | 8 | 0.29 | 0.36 | 0.53 | 0.39 |
| | | | 15 | 0.26 | 0.27 | 0.00 | 0.18 |

reduction was higher than that at RH85% over the same temperature treatments when compared to the control (Fig. 4). The difference in head rice rate between the two RH treatments (RH75% vs. RH85%) was greatest at 35 °C and lowest at 37 °C (Fig. 3), suggesting that RH has a pronounced effect on temperature-induced head rice loss at 35 °C. A pronounced difference was found at 33 °C only in cultivars whose head rice rate was dramatically decreased (LY6 and ZLY47, Fig. 4).

R168 was the most heat-tolerant in terms of head rice rate, except for the treatments of 37 °C × RH 75%, with an MFV of >0.70 (Table 3). In addition, R168 showed a stable and

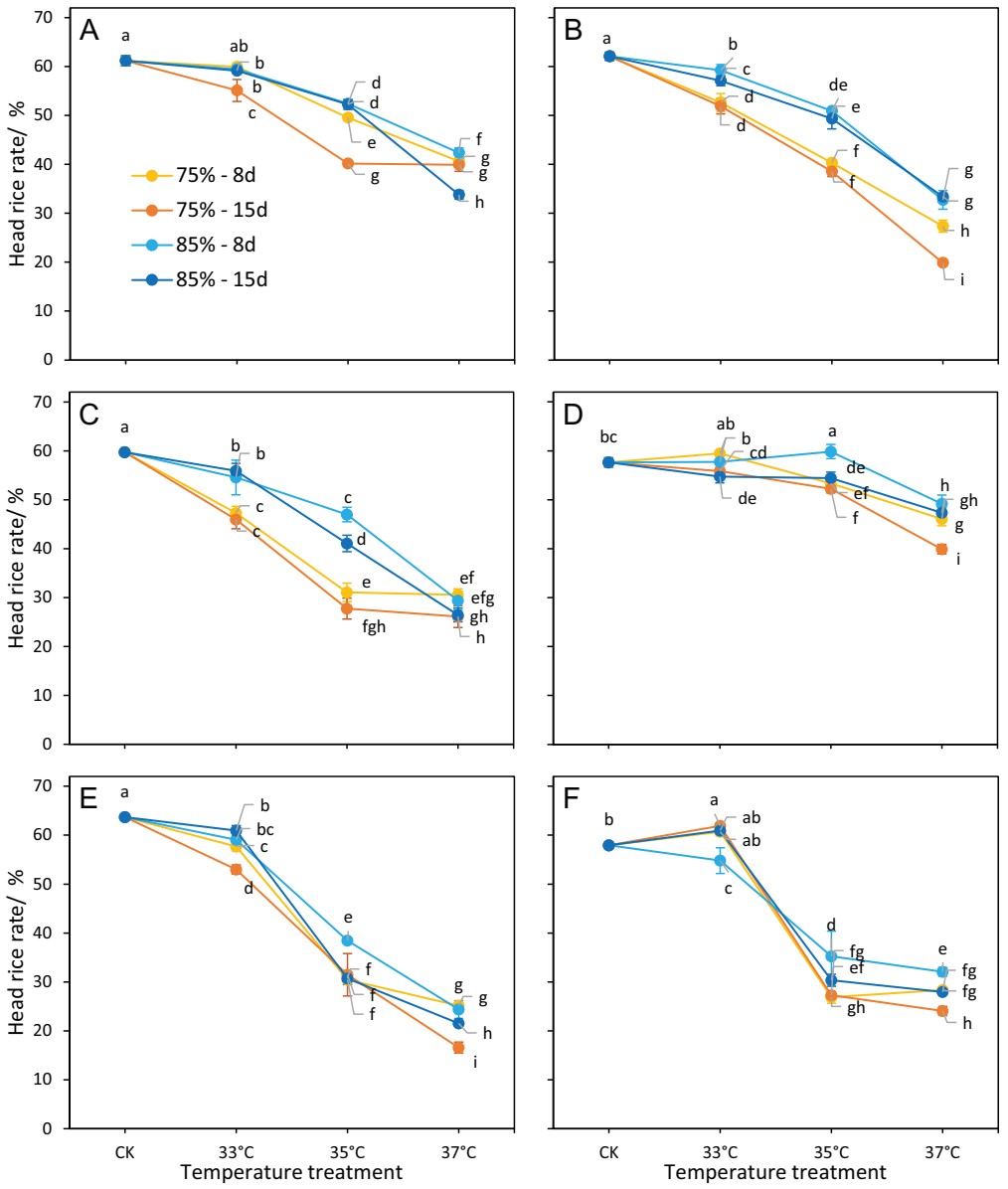

**Figure 4 Head rice rate responses to different treatments.** (A) LY27, (B) LY6, (C) ZLY47, (D) R168, (E) IR64, and (F) 16343. Vertical bars denote standard deviations ($n = 3$), a different lowercase letter indicates significant differences among means of different treatments as determined by the LSD test ($p < 0.05$).

higher head rice rate than other cultivars, even at 37 °C (Fig. 4). The MFVs of head rice rate for 16343 was ≥0.85 at 33 °C but dropped sharply at 35 °C in both humidity treatments (Table 3). For other cultivars, the temperature caused a sharp drop in response to RH. At RH 75% over temperature treatments, the temperature was 33 °C for LY6 and ZLY47, or 35 °C for LY27; under RH85%, the temperature was 35 °C for LY6 and ZLY47, or 37 °C for LY27.

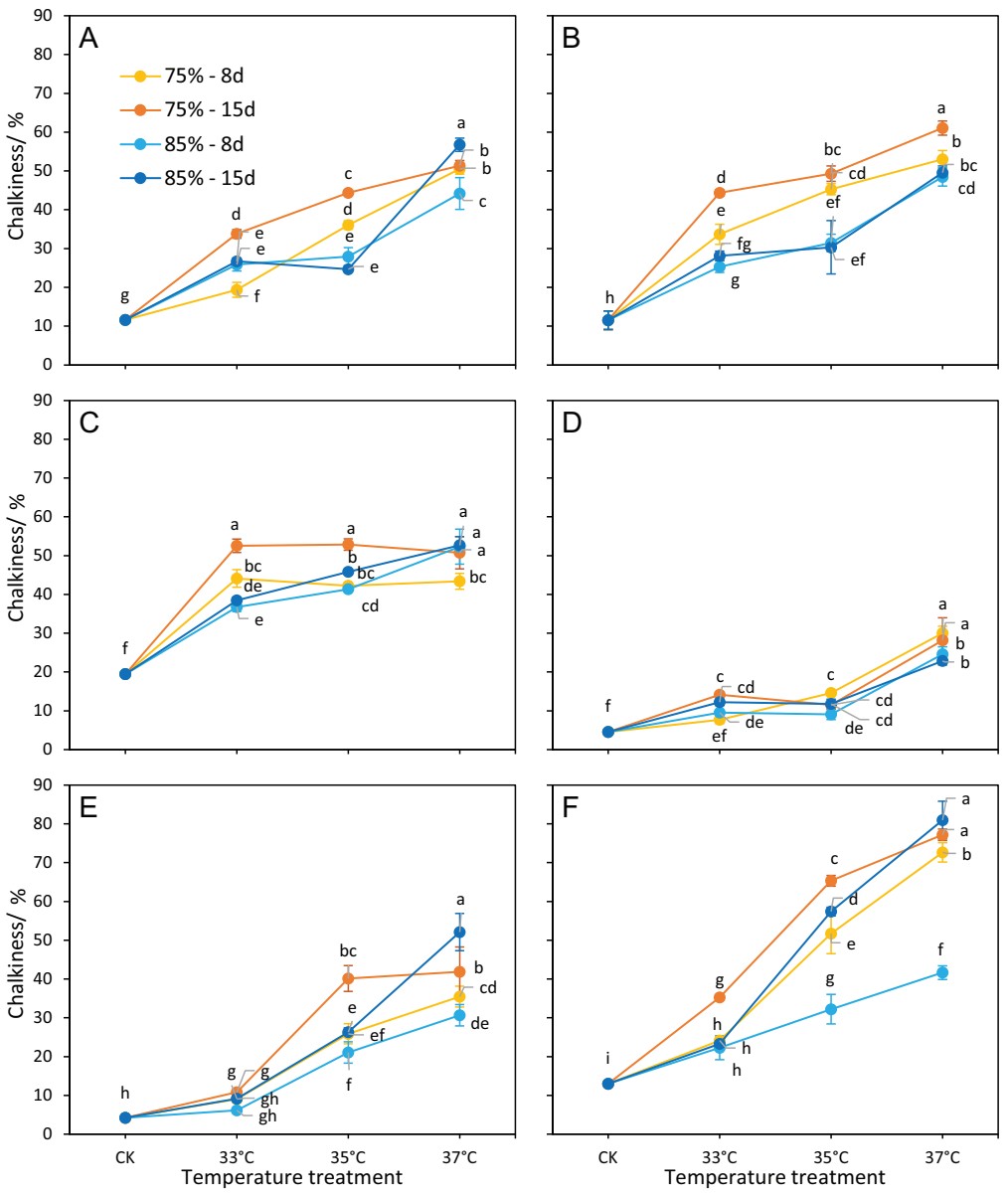

**Figure 5 Chalkiness responses to different treatments.** (A) LY27, (B) LY6, (C) ZLY47, (D) R168, (E) IR64, and (F) 16343. Vertical bars denote standard deviations ($n = 3$), a different lowercase letter indicates significant differences among means of different treatments as determined by the LSD test ($p < 0.05$).

## Chalkiness

For all cultivars, the temperatures, RH, and the interaction of temperature × RH had a significant effect on chalkiness (Table 1). The treatment durations had a significant effect on chalkiness for cultivars except 16,343. The majority of cultivars had higher levels of chalkiness at RH75% than at RH85%, even with the same temperature and duration treatment (Fig. 5). The difference in chalkiness between RH treatments (RH75% vs. RH85%) was most pronounced at 35 °C, followed by 33 °C (Fig. 3). However, the

difference was the smallest at 37 °C. Multiple regression analysis showed that chalkiness was higher when the temperature or durations increased (Table 2).

R168 and IR64 showed the highest MFVs of chalkiness in all treatments (Table 3). R168 also showed the smallest chalkiness difference between RHs (Fig 5). Compared to the control, the chalkiness of R168 sharply increased (more than 10%) only at 37 °C by the RH treatments; this was the case for IR64 at 35 °C and 37 °C, or for the remaining cultivars at all three temperature treatments (Fig. 5).

## DISCUSSION

An upward trend of long-term high-temperature stress coupled with decreased RH during rice grain filling in the Yangtze River basin may occur due to climate change (*Guan et al., 2015*; *Tan & Shen, 2016*; *Zhang et al., 2018b*). Although some studies have shown that hot, dry winds resulted in lower rice yield and a quality loss (*Hiroshi et al., 2012*; *Wada et al., 2014*), RH's sole effect remains unknown. Meteorological factors are interrelated under field conditions; for example, in the Yangtze Valley summer season, which falls between late July and mid-August, the high temperature is typically coupled with low RH, rainy or cloudy days come with high RH (*Guo et al., 2016*). Therefore, it is difficult to separate the effect of humidity on crop yield from the complex meteorological factors in a field experiment (*Yoshida, 1981*; *Zhang, Zhang & Chen, 2017*; *Zhao & Fitzgerald, 2013*). We used controlled environmental conditions to simulate three temperatures with two RH regimes to evaluate RH's effects from those of air temperatures on high temperature-induced rice yield and quality loss during the grain-filling stage. Both temperature and RH showed pronounced effects on grain weight, head rice rate, and chalkiness; temperature × RH combination showed significantly interactive effects on grain weight and chalkiness (Table 1). Temperature treatments at RH75% significantly reduced grain weight, and the reduction was greater than that at RH85%. The same trend was also observed for grain quality loss. Our study is consistent with *Wada et al. (2011)*, who found that grains were losing quality in dry, hot winds.

### The interaction effect of temperature and RH

During the flowering stage, panicle temperature rather than air temperature was curvilinearly related to spikelet fertility (*Weerakoon, Maruyama & Ohba, 2008*). Rice could homeostatically adjust panicle temperature via transpiration cooling for optimal growth. The air-panicle temperature difference was altered under different RH conditions (*Fukuoka, Yoshimoto & Hasegawa, 2012*; *Yoshimoto et al., 2011*). *Yan et al. (2008)* showed that at temperatures of 31.5–33.5 °C, the temperature difference between the air and panicle was about 2 °C under humid atmospheric conditions (~86% RH) or about 5 °C under a dry atmospheric condition (~48% RH). Rice panicles benefit from their transpiration cooling under heat stress at flowering (*Matsui et al., 2014*). However, our results showed decreased RH, coupled with a high temperature at the grain filling stage reduced grain weight and lowered grain quality. The benefit from the decreased RH by transpiration cooling may not be the only scenario of RH and temperature interaction.

Theoretically, there may be a balance between transpiration cooling and a water deficit in rice panicles exposed to heat. *Zhao & Fitzgerald (2013)* reported that a daily maximum temperature ranging from 30 °C to 33 °C and a lower RH led to higher head rice yield and lower chalkiness. Meanwhile, *Wada et al. (2011)* showed dry winds at day/night temperatures of 34/26 °C caused water deficiencies in panicles and restricted starch accumulation, which led to a decline in the quality of the rice. We found that decreased humidity coupled with temperature treatments caused grain weight reduction and quality loss. This effect was most pronounced at 35 °C, and to a lesser extent at 33 °C, and insignificant at 37 °C. This indicates that transpiration cooling may not be enough to compensate for the adverse effects of water deficit when rice is exposed to high-temperature stress (35 °C and 33 °C).

## Physiological mechanism of temperature and RH interaction

Water fluctuations for growth and transpiration are linearly superimposed (*Nonami & Hossain, 2010*), and the impaired ability of stomatal regulation of rice spikelets has a greater evaporative demand under high temperature (*Garrity, Vidal & O'Toole, 1986*), leaving the spikelets and grains at risk of water deficit (*Tanaka & Matsushima, 1971*). Water deficiency in the panicle caused by dry, hot winds was detrimental to rice grain weight and quality formation (*Hiroshi et al., 2012*; *Kang et al., 2003*; *Wada et al., 2011*). Low relative humidity may reduce the assimilation capacity of leaf photosynthesis (*Tanaka & Matsushima, 1971*). The normal maturing process of grains is dependent on a sufficient water supply (*Cochrane, Paterson & Gould, 2000*; *Ferrise, Bindi & Martre, 2015*). Thus, a higher temperature with lower humidity may induce high temperature-forced maturity.

Panicle water potential is temporarily reduced during periods of dry, hot wind stress (*Wada et al., 2011*). The osmotic adjustment of endosperm cells with increased transport was activated to maintain kernel growth, but starch biosynthesis was slowed (*Wada et al., 2014*). The vacuolar structures in the cytosol were preserved during maturity because of osmotic adjustments, resulting in ring-shaped chalkiness (*Hatakeyama et al., 2018*). High-temperature stress led to substantial solutes to be accumulated in endosperm cells by osmotic adjustment. This was accompanied by the partial inhibition of amyloplast development and the formation of protein bodies, which caused air spaces to remain in endosperm cells during grain dehydration, resulting in a chalky appearance (*Wada et al., 2019*). Similar mechanisms of starch synthesis restriction were found under high temperature and dry wind conditions, meaning that decreased RH may aggravate the high-temperature effect on grain weight and quality loss.

The osmotic regulation ability of plants is closely related to the heat tolerance during the vegetative growth period (*An, Zhou & Liang, 2014*; *Jiang & Huang, 2001*; *Zhou et al., 2018*; *Zou et al., 2016*), and the study of *Wada et al. (2014)* suggests that this relationship is also applicable at the filling stage of rice. We speculate that the differences in variety performance in this study may be related to osmotic regulation, but further study is needed.

### Varietal differences under RH in combination with temperature treatments

A daily mean air temperature over 25 °C during the grain filling stage can cause a loss of quality in the rice grains (*Morita, Wada & Matsue, 2016*; *Wu, Chang & Lur, 2016*). However, the magnitude of heat stress-induced damage varied by genotype. *Cooper, Siebenmorgen & Counce (2008)* reported a reduction in grain quality in susceptible cultivars at nighttime temperatures of 20 °C, but this effect was not reflected in heat-tolerant cultivars at the nighttime temperature of 30 °C. We found that grain weight and quality traits did not change linearly with increasing temperatures and only changed dramatically at a certain temperature, with varied responses among different cultivars (Figs. 2, 4, and 5). This was similar to the response of spikelet fertility to high temperatures during flowering (*Jagadish, Craufurd & Wheeler, 2007*).

The identification of heat-tolerant rice germplasms (e.g., N22) may present opportunities to breed heat-tolerant rice at the flowering stage (*González-Schain et al., 2015*; *Tetsuo & Shouichi, 1978*). The mechanism leading to grain quality losses is more complex than the mechanism of heat-induced yield loss (*Jagadish, Murty & Quick, 2015*), which brings more challenges for screening and identifying tolerant varieties. Previous studies paid closer attention to temperatures in controlled environments when investigating rice heat tolerance at the grain filling stage (*Chen et al., 2017*; *Shiraya et al., 2015*; *Tanamachi et al., 2016*). We showed that changes in grain weight and quality are affected by interactions between temperature and humidity and found that humidity is important in evaluating varietal heat tolerance. We suggest that the effect of humidity should be considered in multi-variety tolerance screening and identification. The optimum combination of 35 °C by RH75% by 15 days should be recommended to screen for heat tolerance of rice. Our results bring attention to the detrimental interactive effects of high temperature and humidity on rice yield and quality and are of interest to breeders and agronomists to adjust breeding targets. R168, the most heat-tolerant cultivar used in this study, showed smaller differences in grain weight and quality between two RH regimes. This variety may be selected as a heat-tolerance variety to improve rice yield and quality under climate change.

## CONCLUSION

We found that decreased RH aggravated the detrimental effects of high temperature on grain weight and quality. These effects were the most pronounced at 35 °C, less pronounced at 33 °C, and were not significant at 37 °C. Heat tolerant cultivars were identified and determined to be less affected by the treatments.

## ACKNOWLEDGEMENTS

We thank Yizhi Bao, Xiangwen Li, and Qilin Mu for their technical assistance in the field management.

## Funding

This work was funded by the National Key Research and Development Program of China (2016YFD0300108, 2018YFD0301306 and 2017YFD0301405), the State Key Laboratory of Hybrid Rice, and the Project Center of the Engineering Research Center of Ecology and Agricultural Use of Wetland, Ministry of Education, Yangtze University. The funders had no role in study design, data collection and analysis, decision to publish, or preparation of the manuscript.

## Grant Disclosures

The following grant information was disclosed by the authors:
National Key Research and Development Program of China: 2016YFD0300108, 2018YFD0301306 and 2017YFD0301405.
Engineering Research Center of Ecology and Agricultural Use of Wetland, Ministry of Education, Yangtze University.

## Competing Interests

The authors declare that they have no competing interests.

## Author Contributions

- Haoliang Yan conceived and designed the experiments, performed the experiments, analyzed the data, prepared figures and/or tables, authored or reviewed drafts of the paper, and approved the final draft.
- Chunhu Wang performed the experiments, analyzed the data, prepared figures and/or tables, and approved the final draft.
- Ke Liu analyzed the data, authored or reviewed drafts of the paper, and approved the final draft.
- Xiaohai Tian conceived and designed the experiments, authored or reviewed drafts of the paper, and approved the final draft.

## Data Availability

The raw data and R script used for statistical analysis and figure preparation are available as Supplemental Files.

## Supplemental Information

Supplemental information for this article can be found online at http://dx.doi.org/10.7717/peerj.11218#supplemental-information.

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
