# Peer review of "Detrimental effects of heat stress on grain weight and quality in rice (Oryza sativa L.) are aggravated by decreased relative humidity"

_PeerJ, doi:10.7717/peerj.11218_

## Round 0.1 · original submission · Major Revisions

You have prepared an interesting paper describing an interesting dataset. However the reviewers have identified multiple issues with the manuscript. They have provided a list of suggestions. Please address all of them. Pay special attention to statistical analysis, the study design is not obvious from the description.

Reviewer 1 ·

Basic reporting

Yan et al.’s manuscript tested the effect of heat-relative humidity coupling in 6 cultivars of rice and found that low RH and long duration of stress after anthesis generally led to reduced grain weight, head rice rate and chalkiness. Yan et al. also identified that some cultivars appeared to be more heat tolerant than the others. Yan et al. argued that this body of knowledge will prepare for better agricultural practice in the changing climate.

Abstract

The abstract contains too many results (especially those values, e.g. 0.65 g, 1.68 g), which compromise its readability. More context should be given at the beginning to describe the hypotheses and objectives of the study. Also, abstract should contain a few sentences on the implications of the results at the end.

Introduction

l. 45: As a non-specialist of agricultural crops, I propose that a little explanation of ‘grain filling period’ could provide a better context for the study. Is this a significant period in rice’s life history? Why does high temperature impact the grain filling period in particular? What mechanism?

l. 83: Throughout the entire introduction you did not mention anything about the varietal differences, but they are put as one key objective of the study. I suggest the authors to discuss how varietal differences arise, and hypothesise why they may respond to heat-RH differently.

For the introduction part, now lots of focus have been put on the results and values from different previous experiments, which are very niche. Little insight has been provided on the mechanisms, ecology, physiology of these heat responses. Heat-RH coupling seems an interesting focus but again I would suggest the authors to provider a broader context to discuss how heat and RH interact physiologically in rice or any related crops.

Experimental design

Methods

l. 88: Could you please cite the previous studies?

l. 115 – 117: Please describe in details how milling quality and chalkiness were measured to ensure reproducibility of this study. The measurement framework (National Standard of China for high quality paddy) is not available in the bibliography.

l. 120: Please specify the statistical analyses and the formula in your methods section (multiple linear regression analysis?). Also see below my comments on pairwise comparison test shown in your Figures 2 - 5.

Validity of the findings

Result

l. 126: The durations “of” treatments…

l. 135: “exacerbating”?

l. 137 – 141, 155 – 160, 173 – 176: Why is cultivar not assigned as a fixed factor in the multiple regression analysis? What statistical test has been done to show the variation of heat tolerance among cultivars?

l. 149: which “were” higher

l. 181: missing “.”

l. 181 – 183: This sentence is not clear to me. You meant although some studies have already shown that “hot wind with lower RH …”, the “sole effect” of RH remains unknown?

l. 184 – 185: This temperature-RH coupling effect can be region-specific. Please cite a relevant reference.

l. 191: “grain”

l. 205: “severe grain weight” is unclear to me. You meant “severely decreased grain weight”?

l. 206: “scenario” may be a better word choice than “mechanism”.

l. 235 – l. 238: Quite redundant with the results section with not many new insights. What are the genetic differences among these cultivars? Are they associated with a particular ecoregion? Do they have potential differential adaptation?

l. 242 – l. 245: I feel like this part is important as the authors try to link the results in this study (heat tolerance among cultivars) to agricultural implications under climate change, however it is not fully elaborated. For example, what screening methods can be done? Any agricultural systems have already adopted similar screening methods?
Conclusion

l. 253: “less prominent” may be a better word choice than “finite”.

Figures & Tables

Figures 2 – 5: The figures look to me quite confusing as trends are not clear. I would suggest using line graphs instead of bar graphs: x-axis as temperature, one line for each duration * RH combination (4 lines). Or any other format that may help improve the readability (e.g. split panels between durations / RH).

Figures 2 – 5: You haven’t mentioned any pairwise comparison in the statistical methods, yet these figures showed such. Please describe all statistical procedures you used in the methods section.

Figure 3: The y-axis labels are not clear. The x-axis labels are also missing some characters. Also, why can the grain weight be negative? Does this graph present the absolute difference between control and treatment groups? The figure legend needs more clarification.

Table 2 legend: The formula lacks a mean (intercept) term μ. Also, this formula should be described in the methods section. Also, for X4, I am not sure what you meant by the “value” of each cultivars.

Additional comments

Summary

The study and experimental design are generally robust and valid.

Some parts in the discussion are redundant with the results section. Some parts in the discussion are only comparing the results in this study with those in other similar studies without discussing more on the stress response in rice. For example, how do these results fit in the current understanding of rice physiology and response to heat / RH stresses? Also, although genotype/cultivar was included as a factor in this study, little insight is provided on why genotypes respond differently. How are these genotypes selected and cultivated? Are they ecotypes from different regions and potentially show different adaptative potentials? Finally, the implication of this study is not fully elaborated. How can we use this understanding to transform the agricultural practice?

There are some minor grammatical mistakes throughout the manuscript, I may have missed some and not mentioned above. Also, the authors tend to use long sentences which sometimes compromise the conciseness.

·

Basic reporting

The language used was difficult to understand. The manuscript was littered with grammatical errors and misuse of punctuation. Sentences were unclear. Even the title was not grammatically correct. It requires “is” to be inserted before the word “aggravated” to make sense. The very first paragraph was incomprehensible with commas used instead of full stops in a number of occasions.

There were a number of occasions where references were not appropriately cited. Two such examples in the introduction are in Ln 41-43 (which cites a review by Jagadish et al 2015 for yield loses of 10% per degree increase in whole season minimum temperature) and Ln 73-77 (that cites Matsui et al 2020 for study conducted in China).

There was no clear hypothesis. In Ln 80 the authors state “To verify this hypothesis, experiments were conducted in a controlled environment with six rice cultivars varying in their heat stress tolerance using 12 treatments.” I could not quite make out what the hypothesis of the study is.

Experimental design

The were fundamental issues with the experimental design which I think the authors should consider in future. First of all is the lack of replication – neither the treatments nor the experiment was replicated. Six chambers were used to impose two treatments. The effects recorded might have been due to differences between the chambers. Comparisons between plants treated to the any of temperature, RH or duration combinations cannot really be compared with a control considering the control plants were in a different growth condition. Control plants were kept under natural conditions while treated plants were in growth chambers. The difference in light, temperature, RH, VPD or any variety of environmental factor between the natural condition and the chambers could have been the reason for the responses reported.

Ln 97-98 the authors report that tillers were cut off. I am surprised by this. Why was this done? How would the authors account for the wounding effect of tiller removal on plant responses?

Ln 101-105 and Fig 5 the authors present some data on the growth chamber temperature and RH. I would have expected some measure of error around those means to be also provided. We know most growth chambers are not that fine-tuned to always give the exact settings they are programmed to. This information is especially important for the RH which is even harder to control to within 10%. I wonder how the 75 and 85% RH treatments were achieved.

The authors provided no actual information on how the data was statistically analyzed. Ln 120-121 covers packages used but nothing else. How was the data analyzed? In the absence of true replicates how were differences between treatments tested?

On another point it would be good if the authors provided some supporting evidence for their claim that the six genotypes were of varying heat tolerance (Ln 87-89). Please cite article(s) with such proof or provide additional data from your lab to support this claim.

Validity of the findings

It is difficult to establish if the results reported here are valid considering the many issues with the experimental design. Responses reported could have been due to many other factors other than the treatments imposed.

Additional comments

I have reviewed your manuscript and appreciate the effort you have put into conducting this work. I must admit that I found it difficult to understand as it was poorly written. Of what I could understand I would say the subject you aimed to tackle is of interest to the scientific community.

Reviewer 3 ·

Basic reporting

This is an interesting study and the authors have collected a unique dataset with applying various treatment. The paper is not well written and designed. However, in my opinion the paper needs to have major revision, after that only it can think for reconsider for publication.

Experimental design

1. Please provide upland variety for the deficit heat stress capacity of these varieties that are than the mechanism.

2. At that stage the grain was harvested what was the moisture content at harvest.
3. The storage life and contribute to the HR. what’s the base of 3 month storage.

4. What statistical design was followed?
RH level =2
Duration=2
Temp=3
Total =12 treatment calibration
Variety=6
Total number of halvet calibration 12x6=72, what is the number of replication?

5. How many replication was there for calibration?

6. How these treatments were fitted to factorial design in order to extract the interaction effect.

7. These experiments seem to have 3 way interaction. What about Temp x humidity x duration x variety?

Validity of the findings

no comment

Additional comments

Dear Authors,

This is an interesting study and the authors have collected a unique dataset with applying various treatment. The paper is not well written and designed. However, in my opinion the paper needs to have major revision, after that only it can think for reconsider for publication.

Comments can be looked into it.
Introduction:
Line 57: Lower humidity causes transpiration cooling during flowering.
Line 73-76: Daily mean temp 300C (with RH decreasing from 85% appropriately 70%) lasting for more than 3 days during the flowering period, led to significant seed-set losses in rice plants.
This statement of line 73-76 and the statement at line 57 is misleading & conflict to each other. Clarify the above both statement and justify to the findings.
Materials & Methods
1. Please provide upland variety for the deficit heat stress capacity of these varieties that are than the mechanism.

2. At that stage the grain was harvested what was the moisture content at harvest.
3. The storage life and contribute to the HR. what’s the base of 3 month storage.

4. What statistical design was followed?
RH level =2
Duration=2
Temp=3
Total =12 treatment calibration
Variety=6
Total number of halvet calibration 12x6=72, what is the number of replication?

5. How many replication was there for calibration?

6. How these treatments were fitted to factorial design in order to extract the interaction effect.

7. These experiments seem to have 3 way interaction. What about Temp x humidity x duration x variety?

---

## Round 0.2 · Major Revisions

You did not fully address the comments and concerns of all reviewers. Therefore, I have to recommend that you have one more round of reviews. Please use the suggestions from the previous round to make the changes.

Reviewer 1 ·

Basic reporting

Abstract

The abstract sees improvement in the structure and clarity after the revision.

l. 29: “Interaction effect” is the correct statistical term.

Introduction

The flow is now clear: rice is important > climate is changing > temperature and RH can affect rice productivity > we need to study temperature-RH coupling and varietal difference.

However, I still have some difficulties in following the structure of this section. In many paragraphs the topical sentence is not relevant to the paragraph body. Concluding sentences are lacking in some paragraphs. I would encourage the authors to work around how the paragraphs break.

Experimental design

Methods

l. 108: Since this is unpublished data, it would be worth to explain and elaborate a bit how they show different heat tolerance?

l. 180: “respectively”?

l. 181: You probably meant “days since start of treatment”.

l. 182: I don’t fully follow what X4 is. What is the difference between X4 and Y?

Can you please link clearer how the statistical methods are testing the two objectives in l.102 – 104?

I see that some other reviewers have serious concerns on the experimental design. I wonder if a conceptual diagram of the experimental design may help.

Validity of the findings

Discussion

I recommend the authors to consider using subheadings to break down the discussion.

I see how the authors discussed temp-RH coupling affected the plant's physiology. But since the study measures grain weight, head rice rate, and chalkiness in particular. I would be particularly expecting the authors to discuss how temp-RH led to variation in these physiological responses. I realise the authors may have discussed these in different parts of the discussion but it lacks some coherence.

l. 331: Wrt QTL and candidate genes, it seems a bit far-fetched. Please elaborate clearly the route how this paper may encourage future studies on genetic screening.

Additional comments

I feel the manuscript has significantly improved and does address the reviewers' comments from the first round of peer review. I do not have any significant concern on the methodology or the validity of the experiment and findings. However, I believe the manuscript could still use some work in the story-telling and presentation as it is still not easily readable. I encourage the authors to think first what each section serves and its flow, then work around the paragraphs to structure your ideas. In particular, the discussion lacks coherence with the objectives of the study.

·

Basic reporting

No comment

Experimental design

No comment

Validity of the findings

No comment

Additional comments

No comment

---

## Round 0.3 · Minor Revisions

You have substantially improved the manuscript. I would like to ask you to proofread the paper, make sure that it is grammatically correct.

Reviewer 1 ·

Basic reporting

I acknowledge the authors have revised substantially to my and other reviewers' suggestion on the presentation of the introduction and discussion.

Experimental design

Methods are satisfactorily revised.

Validity of the findings

Discussion has been satisfactorily revised.

---

## Round 0.4 · Minor Revisions

Unfortunately, the quality of written English is not acceptable. I have made several suggestions, but the corrections are not comprehensive enough. I encourage you to address this issue seriously. The paper is very interesting, and it can be popular if it is properly re-written.

---

## Round 0.5 · Minor Revisions

There are several points that need to be addressed before the manuscript can be accepted.

First, the quality of written English. It was significantly improved, but more work needs to be done.

Second, the validation of the regression model would be a good addition to the paper. Consider dividing your samples into training and testing sets for this task.

And the final point - if genetic differences between the studied cultivars are known, and since genes involved in osmotic stress response and yield are well characterized in rice, adding this to the discussion would be good.

---

## Round 0.6 · Minor Revisions

I still have concerns about the regression. First of all, the model selection is not described. In the methods section, you wrote that:
"The multiple linear regression formula: Y= a*X1+b*X2+c*X3+d*X4+e was used in which Y represents grain weight, head rice rate and chalkiness in each treatment, respectively; X1 represents the max daily temperature in each treatment; X2 represents the relative humidity in each treatment; X3 represents the treatment duration days in each treatment; and X4 represents the grain weight, head rice rate and chalkiness of each cultivar under controlled conditions to eliminate the differences in the variety; with e representing the intercept." However, table 1 contains interaction terms. Therefore, the model is not what has been described. In addition, it is a good practice to select the terms of the regression. Two common strategies are forward selection and backward elimination.

Backward elimination:
1. Start with a full model (including all candidate explanatory variables and all candidate interactions)
2. Remove one variable at a time, and select the model with the highest adjusted R2
3. Continue until adjusted R2 does not increase

Forward selection
1. Start with an empty model
2 Add one variable (or interaction effect) at a time and select the model with the highest adjusted R2
3. Continue until adjusted R2 does not increase

If you are using R, there are packages for model selection that you may want to consider.

The second thing to fix. Please make sure that your punctuation marks are correctly placed. You frequently omit commas before the final "and" in the list.

---

## Round 0.7 · accepted · Accept

You have explained the regression in detail in the Methods section and have noticeably improved the writing. Therefore, I recommend this manuscript for publication.